# Probabilistic Learning and Psychological Similarity

**DOI:** 10.3390/e25101407

**Published:** 2023-09-30

**Authors:** Nina Poth

**Affiliations:** 1Department of Philosophy, Berlin School of Mind & Brain, Humboldt University Berlin, 10099 Berlin, Germany; nina.laura.poth@hu-berlin.de; 2Research Cluster of Excellence, Science of Intelligence, 10587 Berlin, Germany

**Keywords:** probabilistic models, few-shot learning, concept learning, psychological similarity, similarity space, structural resemblance, psychological explanation

## Abstract

The notions of psychological similarity and probabilistic learning are key posits in cognitive, computational, and developmental psychology and in machine learning. However, their explanatory relationship is rarely made explicit within and across these research fields. This opinionated review critically evaluates how these notions can mutually inform each other within computational cognitive science. Using probabilistic models of concept learning as a case study, I argue that two notions of psychological similarity offer important normative constraints to guide modelers’ interpretations of representational primitives. In particular, the two notions furnish probabilistic models of cognition with meaningful interpretations of what the associated subjective probabilities in the model represent and how they attach to experiences from which the agent learns. Similarity representations thereby provide probabilistic models with cognitive, as opposed to purely mathematical, content.

## 1. Introduction

Philosophers and psychologists interested in understanding learning from experience in minds and machines commonly deal with two key notions when formally modeling this capacity: the notion of probabilistic inference and the notion of similarity representation. That these notions somehow inform each other is a historical claim widely known (but rarely defended) at least since Hume, who famously advocated similarity as a tool to *ground* inference in his Enquiry Concerning Human Understanding: “In reality, all arguments from experience are founded on the similarity which we discover among natural objects, and by which we are induced to expect effects similar to those which we have found to follow from such objects. […] From causes which appear similar we expect similar effects” [1] (p. 26).

More recently, the cognitive psychologists Matthew Weber and Daniel Osherson ([2], see also [3]) investigate this relationship with a formal model that derives, for a predicate *Q* and objects a,b, the probability for Qa and Qb as a function of the similarity between *a* and *b*. Accordingly, inferring that *b* has *Q* if one knows that *a* has *Q*, which is expressed by the subjective conditional probability pr(Qb|Qa), is justified (carries inductive strength) to the extent that *a* and *b* are similar. According to this model, a learner is considered ‘rational’ when she assigns a relatively high degree of belief (subjective conditional probability) to the conditional only if her subjective measure of similarity is also relatively high.

The model by Weber and Osherson essentially formalizes an idea that had been expressed earlier in the works of Roger Shepard [4], which culminated in Shepard’s ‘universal law of generalization’ [5]. Shepard’s universal law of generalization says that the likelihood of an agent to generalize a behavior from a known entity to an unknown entity is an exponentially decreasing function of their psychological dissimilarity. To derive this statement, Shepard combined a Bayesian approach to learning with a geometric model of similarity representation (for recent work on this law at intersections between psychology and machine learning, see [6,7,8,9,10]).

The key point of this tradition of research on probabilistic learning and reasoning in psychology is that the perceptual experience of a set of objects as being more or less similar in a set of respects plays a rational, guiding, or justificatory role for the agent in generalizing a given property from one object to the other. In looking back to this psychological tradition, there seems to be an implicit assumption underlying this research that the subjective conditional probabilities expressed in these models cannot do by themselves the explanatory work that cognitive psychologists are seeking. In and of themselves, probabilistic models as mathematical models require additional assumptions; in this case, the suggestion is to use assumptions about psychological similarity and the structure of experience to explain the cognitive rationality of learning.

This paper aims to spell out some of the positive reasons for taking seriously such a foundational perspective on similarity in cognitive science when interpreting probabilistic learning in minds and machines. Additionally, the paper aims to bring different perspectives on similarity representations and probabilistic learning in computational psychology and philosophy of mind and of science closer together. In focusing on a case study of concept learning as a Bayesian inference, I illustrate that the accompanying psychological explanation of why the behavior is rational rests on implicit assumptions about the contents of probabilities. These assumptions should be made explicit in both psychology and machine learning to achieve a greater understanding about how learning from experience works.

I suggest two ways in which these assumptions can be spelled out, one referring to the logic of internal psychological similarity representations, and the other referring to the structural resemblances for an argument of how probabilistic learning and reasoning guides action and environmental control. I illustrate how these explicit assumptions provide essential constraints on understanding probabilistic learning from experience.

The structure of the paper is as follows. Section 2 introduces the explanatory aims associated with probabilistic learning models. While focusing on an example from Bayesian concept learning, I argue that such probabilistic models of learning lack cognitive (representational, semantic) content aside from their mathematical contents. Section 3 suggests to fill out probabilistic computations by appealing to two notions of psychological similarity: similarity spaces and structural resemblance. In Section 4, I point to three important areas for future research, and I end in Section 5 with a brief conclusion.

## 2. Probabilistic Models of Cognition and Functional Explanation in Psychology: The Case of Concept Learning

The key purpose of using probabilistic models in psychology is to provide functional explanations of a target cognitive capacity [11]. A probabilistic model as such serves to extract the functional properties of the target system that illustrates this capacity. In the current context, the relevant capacity is the system’s ability to learn from experience.

Take, as a prominent example, probabilistic models of human concept learning as advocated by Josh Tenenbaum and colleagues [6,12,13,14]. These models trace back to a tradition of computational approaches to concept learning and categorization in cognitive psychology, such as Roger Shepard’s [5] universal law of generalization and John R. Anderson and Michael Matessa’s [15] rational analysis of categorization. As such, these models rely on ‘top-down’, as opposed to ‘bottom-up’ theorizing; they highlight formal and functional descriptions of systems whose behavior satisfies certain rationality constraints, and on this basis they attempt to develop descriptions of their likely algorithmic and material implementations.

This research strategy is typically associated with the scaffold of David Marr’s (1982) celebrated three-levels analysis of vision as an information-processing system (see [16,17] for an early and a recent evaluation). In their recent usage, these models are intended to explain how it is possible for young children and adults to quickly learn the correct extension that a label applies to by seeing only a very few examples of the true concept that underlies the teacher’s use of that label (often referred to as ‘fast mapping’ [18] or ‘few-shot learning’ [19]. (Strictly speaking, learners should infer the intension associated with the concept, as there is no way they could know the true extension. The true extension is a moving target given that the use of the label may also change over time—see [20] for a critical discussion).

The basic task is one of generalization: The agent has to infer the conditional probability that an unknown instance, *y*, falls under a candidate concept, *C*, given that a previously observed example, *x*, falls under *C*. Note that, in philosophy, this problem is known as the *qua problem of indeterminacy*. The categories we encounter exemplify very many properties simultaneously. A Dalmatian is a member of the kind *Dalmatian*; it has black spots and white fur, it is furry, is large, etc. Which of these properties should be learned by the system? This is exactly the problem that our case study of concept learning targets, i.e., inferring which of a host of concepts that possibly apply is most probable. Roughly, modelers represent that task by assuming a set of hypotheses, h∈H, where each *h* pairs a candidate concept with the label, the meaning of which has to be inferred from the limited number of examples, $e_{1},\ldots e_{n}$.

A probability function Pr→[0,1] assigns each hypothesis a value so that the probabilities assigned to hypotheses indexing all candidate concepts mutually constrain each other. Intuitively, if, given what she observes, the agent thinks that “fep” expresses dog rather than mammal, she should probabilistically rank the corresponding hypotheses, so that $Pr\left(h_{<‘fep^{\prime },\mathrm{DOG}>}|e_{1},\ldots ,e_{n}\right)>Pr\left(h_{<‘fep^{\prime },\mathrm{MAMMAL}>}|e_{1},\ldots ,e_{n}\right)$. The model assigns these posterior probabilities following Bayes’ theorem:$$Pr\left(H|E\right)=\frac{Pr(E|H)Pr(H)}{Pr(E)},$$
where the posterior probability, Pr(H|E), of a hypothesis, *H* (which pairs a candidate concept with a label whose meaning is to be inferred), given the available empirical evidence, *E* (i.e., the observed examples), is a function of the product of the likelihood, Pr(E|H), of observing the evidence if the hypothesis was true, and the prior probability, Pr(H), of the hypothesis irrespective of the evidence, taken relative to the probability of the evidence, Pr(E), regardless of the hypothesis.

Learning a concept, in this framework, means changing old posterior probabilities in proportion to the degree of confirmation that a hypothesis obtains from the available examples. How exactly degree of confirmation of a hypothesis given the evidence is measured remains unclear, but [6] initially suggest that it should depend on the ‘size’ of the concept. Thus, they specify the likelihood term in Bayes’ theorem with what they call ‘the size principle’:(1)$$Pr\left(E|H\right)={\left[\frac{1}{size(C)}\right]}^{n},$$
where *size(C*) represents the size of the concept. According to Equation (Equation 1), the tendency to favor narrower concepts strengthens with an increase in the number of examples given for this concept. For example, if a learner is presented with one Dalmatian as an instance for the concept to be paired with ‘fep’, the hypothesis indexing the concept dalmatian should be preferred over alternative hypotheses indexing concepts compatible with that observation, such as dog or mammal. According to the model, with more observations of Dalmatians as instances of ‘fep’, this preference should increase (for a detailed critique of the size principle and implicit assumptions of Tenenbaum and colleagues’ model, see [20,21,22]).

The model has been empirically tested in the context of word learning in young children [14,23,24] and applied to other domains of categorization [25]. Its origins can be traced to Shepard’s [5] and Tenenbaum’s [12,13] earlier results on Bayesian modeling of generalization and concept learning.

With this approach, Tenenbaum and Griffiths offered an alternative to earlier attempts to solve the *qua problem*. These earlier attempts failed due to their exclusive reliance on the notion of Shannon information: A Shannon-information theoretic account is insufficient to individuate the candidate concepts relevant to learn for the system because informational relations are ubiquitous in nature. dog and Dalmatian both correlate with features of Dalmatians, so correlation does not on its own deliver determinate contents to solve the qua problem. Additionally, a disjunction of features is always more likely than each of the conditions individually. For instance, a mushroom may be edible, but it is more probable that it is edible or poisonous. Thus, additional biases are needed to select salient correlations to determine the right concepts to be inferred.

The size principle can act as one such bias in the form of assumptions about how examples have been sampled from the world, or from the category that generates the examples (see [26] for illustrations using systematic random sampling from an urn). Intuitively, observing three Dalmatians as examples for the meaning of ‘fep’ becomes more likely if these examples were in fact random samples from the concept Dalmatian, for if they were randomly sampled from dog, one would expect also other kinds of dogs to to occur in the data. It would be a “suspicious coincidence” [23] (p. 289) if one was to observe three Dalmatians if ‘fep’ in fact meant dog.

From the perspective of psychological science, the model explains few-shot learning by identifying functionally relevant factors that allow the system to adequately map instances of a label (e.g., observations of Dalmatians) to the concept that they belong to (e.g., Dalmatian). Unfortunately, Tenenbaum and Griffiths do not explicate what determines the adequacy of this mapping.

To make this intuition more poignant, we can look to the history of philosophy of science, where we find a strong analogy to Bar-Hillel and Carnap’s [27] ‘inverse relationship principle’. According to this principle, the information value of a proposition is inversely proportional to the probability associated with that proposition. This formulation is analogous to the size principle if ‘proposition’ is taken to mean ‘concept’ and information value is measured by the number of possible things that fall under the concept (i.e., the concept’s intension).

The analogy helps to explicate the point of the size principle: information value indicates how well a hypothesis can be confirmed by a limited set of random observations of instances of the concept. A hypothesis indexing a narrow concept has a low a priori probability. For example, all else being equal, it seems less probable that a praying mantis will fly into my visual field than that any random insect will. Hence, $Pr\left(h_{P}\right)<Pr\left(h_{I}\right)$. However, were a praying mantis to fly into my visual field now, this observation would confer a higher degree of confirmation on $h_{P}$ than on $h_{I}$ (even though both hypotheses are compatible with the observation), and so $Pr\left(e|h_{P}\right)>Pr\left(e|h_{I}\right)$. According to the inverse relationship principle, instances of a hypothesis with a low a priori probability provide a high degree of confirmational import for that hypothesis. In the Xu and Tenenbaum case study, it is assumed that concepts with a narrower intension (e.g., Dalmatian as opposed to dog) are a priori less probable, but for this reason much better confirmed by limited experience. Narrower concepts are more informative and should be preferred as they reduce uncertainty about the true category given limited experience.

To summarize, the case study extracts some functionally relevant factors associated with learning from limited experience: (1) in science, rational learners assign greater probabilities to hypotheses that are better confirmed; (2) how well a hypothesis is confirmed by limited data depends on the information value of that hypothesis; and (3) information value is measured by the semantic content of the hypothesis, i.e., by the ‘size’ of the concept indexed by the hypothesis. However, this explanation is essentially incomplete: If what the hypothesis says is partly constituted by the size of the concept (i.e., the semantic content of the hypothesis), which seems to depend on the number of possible things falling under the concept, how should a rational agent access and represent this information? (The analogous problem for philosophy of science is is to explain how researchers can identify the scope of a scientific hypothesis and from there derive its associated probability).

### 2.1. What Is Missing

In the psychological model, probabilities play the role of attitudes that are associated with beliefs. They have formal, mathematical features but only in combination with their associated beliefs do they obtain cognitive or semantic features. The previous discussion suggests that how much probability a belief becomes assigned should depend on the semantic content of the belief (i.e., the concept). In this section, I want to highlight two issues that are insufficiently addressed in the Bayesian model: (1) based on what principles the hypothesis space should be carved up so that hypotheses obtain the semantic contents they have, and (2) based on what principles that content changes with experience.

To grasp the first issue, recall that, in Xu and Tenenbaum’s model, concepts are defined extensionally, that is, by the things that fall under the concept. Of course, learners actually cannot know the true extension associated with a concept, for it is impossible to know the true number of things that will fall under Dalmatian. Alternatively, learners could represent the concept’s intension, by representing the possible ways in which Dalmatians could be instantiated.

This suggestion is pursued in terms of psychological similarity spaces in Section 3.1 below. Xu and Tenenbaum also rely on similarities but in a different way. They provide an analysis of the semantics of complex categories based on their internal taxonomical structure into the subordinate, basic, and superordinate levels, and they carve distinctions between category levels based on adult subjects’ prior objective similarity judgments on a scale from 1 to 9 (but note that this empirically derived taxonomy may not be useful to represent children’s classifications). In this model, the specification of the hypothesis space already delimits where possible category boundaries can be drawn, and, consequently, which candidate concepts can be inferred. (I am grateful to an anonymous reviewer for highlighting this point). For instance, in Xu and Tenenbaum’s [14] model, Dalmatian competes with dog and mammal, but undetached Dalmatian sclices is already excluded from the hypothesis space (see Quine [28] for the original ‘undetached rabbit slices’ example). What candidate concepts are plausible depends in part on how contrast classes are carved up in the taxonomy of candidate concepts, but the mechanism for doing so is yet missing from the Bayesian analysis.

An alternative proposal is offered by [5], who maps generalization probability onto a geometric measure of stimulus dissimilarity, which is constructed using the method of multi-dimensional scaling, see [29,30]. Shepard’s model does not consider taxonomically structured hypotheses spaces but it nevertheless predicts, by using a combination of probabilistic inference and geometric similarity spaces, a wide range of generalization data across stimuli, species, and modalities.

The issue that I am concerned with here is that, despite their formal elegance and great predictive power, what is missing from these probabilistic models is a closer examination of what justifies these modeling methods and what theoretical contributions they make to explaining concept learning. Most importantly, it remains unclear how the representational content of Bayesian hypotheses spaces should be specified and how that content is active in producing behavior (see [31] for philosophical discussion of the explanatory role of mental content). Below in Section 3.1, I discuss how two notions of psychological similarities, also initially discussed by [5,32] but subsequently discarded by [6], may give rise to mental content of hypotheses for the purpose of rationalizing behavioral learning patterns.

To grasp the second issue, note that in Xu and Tenenbaum’s [14] word-learning example, hypotheses are specified relative to a set of available contrast classes. However, the semantic content of the belief (i.e., the concept) is typically not exhausted by relations to other beliefs or contrast classes (e.g., as suggested by proponents of inferential-role semantics [33]). It also depends on how the belief relates to experience. (Analogously, the degree of confirmation of a scientific hypothesis depends on how well the available evidence supports that hypothesis). It is this relationship that is of issue when we ask about why and to what degree a hypothesis indexing a specific concept like Dalmatian is better confirmed by an empirical observation of three Dalmatians than a less specific hypothesis indexing dog.

The problem is that, although Tenenbaum and Griffiths’ [6] account aims to explain concept learning from limited experience, it totally lacks an account of experience, and an account of the relation between the content of the hypothesis, the concept, and the content of the learner’s experience. Following Equation (Equation 1) and the stipulation that learners assume that the data were randomly sampled, the norms governing few-shot learning are purely determined by looking to the internal coherence of the structural, probabilistic relations among learners’ beliefs about concepts (and how they are used to generate instances), as opposed to how well the learner’s experience matches these concepts. Probabilistic computations manipulate beliefs or concepts, but they do not themselves constitute the contents of the beliefs or the contents of concepts. Progress in machine learning has helped immensely to understand how information can be processed in ways that respect principles of probabilistic coherence. Nevertheless, we currently lack a clear idea of how probabilistic computations arise from psychological contents and from experience.

This problem traces back to prominent discussions concerning nativism in the philosophy of mind of the 1980s. Consider that Xu and Tenenbaum’s word-learning model reflects a symbolic approach to the semantics of conceptual structure of the hypothesis space. In later work, students of Tenenbaum expand this approach to what they call a ‘probabilistic language of thought’ [34,35], in which concepts correspond to symbolic representational primitives structured via lambda calculus, which delivers a compositional syntax. Moreover, this work has been expanded to what has recently been called “Bayesian Program Learning”, where “concepts are represented as simple probabilistic programs—that is, probabilistic generative models expressed as structured procedures in an abstract description language” [36] (p. 1333).

The model by [36] illustrates that the standard argument remains highly influential in computational cognitive science. Their generative model of handwritten characters starts from built-in symbolic representational primitives (e.g., simple strokes) that function as sub-parts to be combined into parts of a prototypical handwritten character. This compositional structure allows the model to generate new types of characters by selecting available primitives from a start-up library of simple symbols that are combined and related into complex symbols, which define simple ‘programs’ (i.e., concepts). The generated object templates at the type level are then ‘run’ to generate exemplars at the token level, where each exemplar is interpreted as raw data. This forward model is subsequently used as a hypothesis when evaluating what object template has been used most likely to generate novel encounters of raw data.

Some philosophers of mind (e.g., [37]) have taken these recent developments in Bayesian cognitive science to motivate and justify a revival of Jerry Fodor’s [38] Language of Thought hypothesis. Fodor argued that (1) conceptual thought is systematic; (2) to explain systematicity we need to posit representations with a certain structure; (3) this structure implies that representations are composed of elementary parts (concepts) that are recurring and systematic; and (4) such a system can only be language-like, where, by ‘language-like’, Fodor meant that conceptual thought is propositional, that is, it illustrates predicate–argument structure, and that simple concepts combine compositionally to complex concepts. At the basis of this view lies a commitment to classical computationalism: thinking is computation; computation is (discrete) symbol manipulation; therefore, concepts must be (discrete) symbols in a Language of Thought.

Philosophers of mind have largely disputed premise (4) of this argument. See [39,40,41] for two kinds of insightful alternative perspectives. While Camp claims that cognitive maps can in a sense be associated with predicate–argument structure, Rescorla proposes a combination of a Bayesian and geometric-spaces approach to explain robot navigation. This latter approach is a weaker argument against the need for symbols since geometric structure does not maintain predicate–argument structure; it is similar to the suggested combination of Bayesian inference and Conceptual Spaces in Section 3.1 below.

One of the major issues with Fodor’s symbolicist view was that it ended up in “mad dog nativism” [42]. Fodor denied that primitive concepts, C, (and later also complex concepts) could be learned by hypothesis formation and testing (HF). He argued: “HF assumes that learning C is coming to believe that C applies to Cs. But beliefs are constructs out of concepts, not the other way around; […] HF, when construed as a model for concept learning, reverses the proper order of explanation as between models of concept learning and models of belief acquisition” [43] (p. 139).

More specifically: (1) Concept learning is a process of rational inductive inference (hypotheses formation and testing). For instance, from some observations of trees that are green we conclude: ‘All trees are green’. (2) Testing a hypothesis about a property, P, requires having a concept, C(P), of that property. (3) “If a creature can think of something as green or triangular, then that creature has the concept green or triangular”, and “the C things are the ones that are green or triangular”. (4) Therefore, concepts cannot be learned.

This critique of inductive models of concept learning applies to the Bayesian models just as well, since they assume an innate start-up library of primitive symbols, or, to say it in the words of Rutar et al. [44]: “the total number of hypotheses, latent plus explicit, cannot be changed through learning”. As in Fodor’s case, the problem traces back to the fundamental assumption of the symbolic view that ‘concepts’, ‘hypotheses’, or ‘programs’ are interpreted as propositions, in which inferences range across symbols whose contents are merely stipulated, and it remains unclear why hypotheses express the contents they do.

These issues put into question the explanatory value of principles such as size and inverse relationship for the cognitive and psychological sciences. Although these principles seem plausible to analyze scientific hypotheses, and they can be adjusted to predict children’s word-learning behavior, these ingredients seem insufficient to provide a psychologically plausible account of concept learning, that is, an account that appeals to psychological capacities and the semantic relations between concepts (belief) and experiences (evidence).

The concern that Bayesian rational analyses focus too much on a mathematical characterization of behavior and too little on the identification of cognitive content has been elucidated in much detail in a heated discussion between [45,46,47,48,49]. Probabilistic models have been accused of behaviorism due to their lack of attention and commitment to talk of mental representations. Probabilistic models have been favored over alternatives due to their unificatory power [50,51] but the relationship between unification and (causal) explanation has been questioned across domains in science continually [52]. Generally, these discussions highlight the need for more psychological theorizing in Bayesian models of cognition, as opposed to building more computationally sophisticated models.

Conditional probabilities are used to explain how to change predictions given environmental changes, assuming learners are equipped with the right priors. What they do not provide is (a) information about what external norms guide learning so as to justify evaluating the behavior as ‘adaptive’ to the agent’s environment and (b) an argument as to why probabilistic norms of learning and reasoning give rise to cognitive behavior.

In summary, these issues call for a stronger focus on the semantics of probabilistic inferences rather than on the computational principles based on which they may be processed in the mind. The case study on Bayesian concept learning illustrates a rational analysis of concept learning from limited experience. Learners are rational in placing a higher probability value on hypotheses indexing narrower concepts because such concepts increase the likelihood of observing limited data that has been generated by a systematic random process. The rationale for assigning higher probabilities in this way is information maximization, and the information value of a hypothesis is assumed to depend on its semantic content, that is, the number of possible things falling under the concept indexed by the hypothesis.

What is missing from this computational approach is a psychological theory of how concepts can be individuated in learners’ minds and how they relate to experience. Aside from information maximization, we need to motivate also why these models have cognitive contents. The purpose of stipulating probabilistic computations (in cognitive science) is to explain the system’s behavior (e.g., learning word meanings) and the conditions under which it counts as ‘successful’. A view of probabilistic computation does not provide these explanatory conditions by itself, it merely provides an account of how attitudes associated with the relevant psychological representations change as a result of internal computation and experience, without saying anything about the nature, content, and status of these representations (e.g., beliefs, concepts) so manipulated. Thus, since probabilistic models of cognition by themselves fail to offer an account of probabilistic representation (e.g., concepts), they are at best incomplete as accounts of learning concepts from experience. (For an argument that probabilistic models currently fail to provide an account of mental content in perceptual experience at the subpersonal level of information processing, see [53]).

In the next section, I suggest that researchers in Bayesian cognitive science can and should consult theories of mental representation to explain the psychology of learning from (limited) experience, and I focus on two similarity-oriented perspectives.

## 3. Filling out Probabilistic Computation in Cognition: Two Similarity-Oriented Perspectives

In this section, I discuss one possibility to respond to these worries by suggesting to ‘fill out’ the contents of probabilistic computations by referring to psychological similarity representations. The motivation is to focus more strongly on how hypotheses should be specified to begin with, and how they should be assigned probabilities as a result of experience. In particular, my claim is that subjective probabilities in cognition should be treated as representing similarities among learners’ actual experiences (an aspect of the evidence) and similarities among possible experiences (a partial aspect of the content of beliefs). Scaling experiences according to some measure of similarity becomes a precondition for optimal learning from sparse experience.

This response divides into two further options, one being an appeal to internal psychological similarity representations, while the other appeals to structural resemblances. The aim of this section is not to invent two wholly new solutions to the problem of inductive concept learning, but to bring these solutions into a systematic relationship and show why each option provides valuable constraints on interpretations of representational primitives in probabilistic models of learning.

The first solution specifies content in hypotheses spaces with geometric distances; concepts correspond to regions in geometric space, and it looks to internal constraints that demand that how hypotheses are specified does not violate the metric axioms, and that probabilities are assigned to hypotheses in correspondence with geometrically specified principles of cognitive economy. The second solution proposes to specify the content of hypotheses in terms of second-order homomorphisms. In this case, the norms governing how probabilities are assigned to hypotheses are oriented towards the fit to the external environment statistics.

### 3.1. Similarity Spaces: Internal Magnitudes of Experience and Belief

We continue to assume that probabilities figure in the attitude associated with the system’s mental representations (e.g., concepts; stipulated to explain the system’s behavior), and these attitudes change coherently with the laws of probability. However, we add to this a model of the representational content associated with concepts (i.e., what was yet missing from probabilistic models when used for the purpose of explaining psychological behavior).

Such a model is offered by the Conceptual Spaces framework by Peter [54], which builds on the notion of the psychological similarity space developed by Roger [29,30] and which aims to study concepts and categorization mathematically by looking to geometry and topology. A conceptual space is structured by a set of quality dimensions that measure perceptual qualities [55]. That is, the values that an object is assigned to along the dimensions of conceptual space represent perceived qualities of objects (e.g., brightness, height, time, pitch, taste, etc.). For instance, any example of color perception can be analyzed as a mapping that attributes a specific value to the hue, saturation and brightness dimensions. Each point in this space represents a specific color shade (e.g., $red_{\langle 2,9,1\rangle }$) and the geometric distance between two points represents the perceived dissimilarity associated with the two color shades. (Although the model attempts to represent psychological similarities in individual perception, in practice, the method relies on multidimensional scaling, which typically aggregates data across groups of subjects. Hence, we may more appropriately talk about an idealized model of psychological similarity).

Concepts are modeled as regions in this space; the size of the region indicates the average dissimilarity among color shades located in that region. Accordingly, the region representing red is larger than the region representing coral red, since the average distance among points falling into the former is larger than in the case of the latter.

This similarity-spaces framework acts as a basis to fill out the representational contents of probabilistic computation in cognition and thereby it completes the missing psychology from probabilistic models of concept learning as discussed in Section 2. Psychological similarity representations provide the semantics to individuate concepts indexed by hypotheses.

In contrast to Bayesian Program Learning (Section 2.1), Conceptual Spaces expands on classical symbol-manipulation by assuming that symbols are attached to concepts, which, however, form their own kinds of cognitive representations; concepts themselves are derived from perceptual and sensorimotor experiences. As such, the theory postulates that concepts provide cognitive content to symbols, as opposed to assuming that concepts themselves are symbols. The systematic re-combination of this information in reasoning follows principles of geometric organization, as opposed to an innate syntax. For instance, reasoning about colors will be constrained by the proximity structure of perceptual color space. For reasoners endowed with the CIELAB-color space, it is trivially true that nothing is both red and green all over because red and green do not overlap in that space. (Vision researchers commonly use the CIELAB space to analyze perception of colors shown on paper or cloth, and the CIELUV space to analyze perception of colors shown on a screen).

In postulating the scaffold of geometric quality dimensions, the theory offers an improvement in understanding learning from experience, the idea being that experience is encoded in quality dimensions, which in turn determine the semantic contents of these hypotheses. Furthermore, hypotheses are assigned probabilities on the basis of the ‘size’ of a concept. Under adoption of the similarity framework, the intension of a concept is the area covered by a region in psychological similarity space [22].

It is then possible to understand probabilistic learning from experience in terms of the overlap of how well the candidate concept (e.g., the region red vs. the region coral red) overlap with instances experiences as red (i.e., instances occupying points within these areas in color space). The degree of confirmation of the hypothesis is the overlap of the concept indexed by the hypothesis and the experience, relative to the similarity space. For instance, the hypothesis that the true meaning of a given label is the concept coral red is better confirmed by a few observations of coral red shades than the hypothesis that the meaning is red because coral red overlaps more closely with coral red shades than red in conceptual space. The same goes for the learning of more abstract concepts such as Dalmatian versus dog. Aside from bearing advantages for relating the contents of experience and belief in probabilistic learning accounts, the model bears other advantages. For instance, it captures the gradual nature of perception, sensorimotor intentions, and conceptual thought (see [56,57] for examples).

Recent work in the philosophy of cognitive science has already combined probabilistic models of learning with the similarity spaces framework to improve psychological explanations of perceptual, motor and category learning. Yraissati, Douven, Decock, and other colleagues focus on criteria of naturalness in geometric color and odor spaces to explain under what conditions hypotheses in probabilistic inferences lead to learning that optimally trades accuracy with fastness [58,59,60,61,62].

This said, Gärdenfors’s [54] theory of learning concepts does not favor a probabilistic approach to learning, albeit various applications that show that the theory is compatible with Bayesian learning principles [22,63]. These applications of the theory highlight that how probabilities are assigned depends on what dimensions are salient and also which dimensions are combined into domains. Ref. [63] relies on the similarity-spaces framework to explain how inferences transition between perception, motor control, and belief (and back) from the perspective of a naturalized epistemology. Refs. [20,22] focuses on the unificatory power that probabilistic analyses of learning can offer to render diverse interpretations of the notion of psychological similarity mutually informationally relevant.

What is common to all these approaches is that, to the extent that they explain behavioral patterns that illustrate probabilistic forms of inductive inference (e.g., perceptual color categorization), they stipulate that the relevant inferences range over areas in psychological similarity space, and that it is the geometric structure of that space which grounds probabilistic computation in representational content and experience.

What is the distinct contribution delivered by the geometric-spaces solution? Different answers arise, depending on whether one follows Shepard’s, Tenenbaum and colleagues’, or Gärdenfors and colleagues’ presentations of this approach. First, Shepard’s [5] presentation highlights methodological benefits for modeling empirical data on similarity judgments and making accurate predictions about generalization in many domains. From this perspective, the geometric solution is an elegant and predictively powerful tool to model the problem of how an ideal Bayesian agent should infer the most probable concept that has generated the observed data.

For instance, selection of relevant dimensions in Shepard’s initial work was motivated primarily by methodological aims of finding a parsimonious modeling solution to predict categorization behavior and accommodate experimental data [64]. Likewise, other computational psychologists such as [65] introduce dimension weights to accommodate directionality and context effects (see [66]) that otherwise pose an anomaly to the claim that generalization behavior is guided by geometric similarity representations.

Second, the authors of [6] see it as one of the greatest advantages of their probabilistic approach that Bayesian inference is independent of the geometric-spaces account because, they argue, it presents a solution to the learning or generalization problem at Marr’s [67] computational level of analysis. They see this agnosticism as the key ingredient to Bayesianism’s power to unify Shepard’s geometric and Tversky’s set-theoretic accounts of psychological similarity. (In fact, this unification turns out to unify their behavioral predictions, not their classically opposing accounts of psychological similarity—see [20] for details). Thus, they suggest little commitment to geometric spaces as a specifically preferable solution over alternative ways of modeling the contents of hypotheses.

Third, recent applications of Conceptual Spaces [20,22,61,63,68] go beyond the solution at the computational level and answer to questions concerning how representational primitives should be set up and interpreted in the model and what meaning this set-up carries for justifying the explanatory power of the model. How similarity space is carved up into discrete regions is justified on the grounds of coherence with the axioms of geometry, plus considerations of cognitive design principles.

The authors of [62] propose as such principles parsimony, informativeness, representation, contrast, and learnability, all of which are given a geometric interpretation in their accounts (but see [68] for a critique). In contrast, it is not always clear according to which *cognitive* design principles representational primitives in Tenenbaum and colleagues’ models are selected. For example, the justification for the taxonomic construction of the hypothesis space to model children’s learning of concepts in [14] builds on adults’ similarity-judgment data of training items. This is surprising, since adults have already learned the relevant concepts, in contrast to children.

Gärdenfors also attributes methodological benefits to the geometric solution, but these are theoretical, as opposed to empirical. He characterizes Conceptual Spaces as “a theory that *complements* the symbolic and the connectionist models and *forms a bridge* between these forms of representation” [69] (p. 24, my emphasis). Neuronal representations are subsymbolic; they are structured by weighted connections and dynamic activity patterns of neural networks, resembling the dynamical activations of neuronal representation.

Conceptual Spaces suggests bridging earlier divides between subsymbolic and symbolic approaches by structurally relating representations at these distinct levels of cognitive representation. If patterns of activation at the neural level are regular enough over longer periods of time, stable attractor points can emerge in a hyperdimensional plane that provides the dimensions to structure conceptual space. That is, under sufficient repeated activation, these attractor points “will form a low-dimensional hypersurface in the high-dimensional state space of the [artificial neural network which] can be interpreted as a conceptual space” [54] (p. 245).

At the other end, the symbolic level obtains its content from the conceptual level. In particular, the meaning of a symbol emerges from the conceptual level when the system learns to form associations between concepts as represented in conceptual space and symbols in a database, for instance, when it leans to associate ‘blicket’ at the symbolic level with the concept Dalmatian in conceptual space. The conceptual level explains how the use of labels maps onto differences in the similarity of category members. For instance, ‘blicket’ only applies to things that are more similar to the prototypical Dalmatian than to prototypes of relevant contrast classes (e.g., labrador, husky, corgie) at the conceptual level; this contrast is not expressed in the symbols. What makes representations in conceptual space unique is that only they, not the subsymbolic or symbolic representations, are structured by geometric distances.

Additionally, the theory of Conceptual Spaces [54,70] bears stronger epistemic commitments than the methodological approach pursued by Shepard [5]. From the perspective of Conceptual Spaces, it is a mistake to give up Shepard’s initial commitment to geometric representations; these remain relevant constraints on learners’ inferences in the model. Geometric structure is justified by the assumption that it presents the ideal way to describe psychological properties and the internal logic of learning and reasoning based on the coherence of learners’ internal representations with the metric axioms. As such, the theory of conceptual spaces not only uses the idea of geometric spaces as an instrumental tool to predict generalization behavior, as did earlier mathematical approaches, but the theory carries a stronger commitment to the claim that geometric dimensions in the model encode information about perceptual and sensorimotor information, i.e., that the similarity metric indicates how qualitatively different experiences are.

Finally, in other work, Ref. [32] also endorses stronger metaphysical and epistemic commitments. He assumes that geometric spaces are an optimal way for an agent to represent the world because such spaces ‘mirror’ the world in a manner that guides adaptive behavior. In this work, geometric structure is not only a methodological tool for accommodating data but also taken to be the best hypothesis to explain how agents represent and categorize the world to navigate it successfully. More on this claim follows in the next section. There, I discuss structural resemblance approaches, which postulate external constraints on setting up psychological space.

### 3.2. Structural Resemblance: How to Render Learning from Experience as Action-Guiding

There is another way in which subjective probabilities in cognition may derive from psychological similarities that are worth using to constrain psychological explanations of behavior (e.g., concept learning) based on probabilistic models. The key to grounding probabilistic computation in structural similarities is their action-guiding role.

The authors of [53] explore the possibility that, instead of attaching to beliefs or concepts as a matter of attitude to propositions, probabilities attach as a matter of the *content* of perceptual experience. On this contrastive view, probabilities arise directly from experience; the agent’s interaction with the statistics of the environment is constitutive of the perceptual experience itself. The motivation is to accommodate, in this way, the common intuition that perceptual experience itself represents uncertainty [71,72]. According to this perspective, the probabilistic contents of perceptual experience arise from the ecological properties of the agent’s interaction with its distal environment; probabilistic mental representations are “states that in some sense mark or represent information about the probabilities of distal conditions” [53] (p. 907).

Lee and Orlandi contrast this understanding of probabilistic experience to the classical epistemological notion of degree of belief. Importantly, the evaluation of the accuracy of probabilistic computations is different in this case. A learner is rational in how she adjusts her probabilistic representation of the world if these adjustments correspond, in some action-guiding manner, to changes in the statistical invariances in her environment. That is, the norms of learning are not determined by internal probabilistic coherence but by a mapping between the internal probabilistic states and the environment statistics. Psychological similarity so construed (as correspondence) is relevant for probabilistic learning because, while Bayesian norms of learning ensure internal coherence (i.e., they transition among beliefs or between belief and experience), structural similarity ensures a rational connection to the outside world.

This perspective suggests that changes in probabilistic experience, insofar as they correspond to changes in the environment statistics, need not necessarily obey the laws of probability calculus. They need to satisfy constraints on an appropriate correspondence relation so as to allow the agent to appropriately act in the world.

At this point, readers might object by pointing to Dutch book arguments in favor of probabilism, which show that having coherent credences leads to reasoning that is more action-guiding than having incoherent credences, where actions are represented by bets. I agree with Lee and Orlandi that the scenario of a Dutch book seems too far removed from cases in which experience itself appears to encode uncertainty. A basic assumption in Dutch book arguments is that there is a connection between the thinker’s credences and how much she is willing to pay for a bet on the relevant proposition; if she is more confident in the proposition, then she should accept to pay more on a ticket selling on that proposition. Dutch books show that agents whose reasoning violates the probability axioms are vulnerable to a guaranteed betting loss from a set of bets that are individually accepted as fair by their credences.

The problem with Dutch book arguments is that they only fit cases in which probabilities figure in as part of the attitude and not as part of the experience. An agent may not be willing to pay anything for a bet on any proposition; the agent may not want to enter a bet in the first place. Hajek provides a nice illustration for this: “if you go to Venice, you face the possibility of painful death in Venice; if you do not go to Venice, you do not face this possibility. That is hardly a reason for you to avoid Venice; your appropriate course of action has to be more sensitive to your credences and utilities”, and he concludes: “So violating the probability calculus may not be a practical liability after all. Objections of this kind cast doubt on an overly literal interpretation of the Dutch Book argument” [73] (p. 10). Dutch book arguments are too idealized to treat cases associated with the rationality of experience itself.

As Ref. [74] (p. 599) puts it, recent probabilistic models just assume but do not justify that probabilistic representations are mental entities. What structural resemblance accounts can do for probabilistic models of learning is provide reasons to think that these models have something important to say about the *psychology* of learning and reasoning.

In this usage, structural similarity accounts have played an important role in the philosophical literature on mental representation, which aims to explain the contents of mental states (e.g., thoughts, beliefs, desires, etc.) and how mental states can be about things [75]. Specifically of interest as a background to the discussion in this section are teleofunctional accounts such as [76].

On teleosemantic accounts, the semantic content of a representation depends on a non-semantic relation between the representational vehicles (e.g., neuronal activity) and their targets (e.g., experimental stimuli). Ultimately, the framework assumes that, if a representation can be usefully modeled in a psychological similarity space, this is because of neural or sensorimotor and ultimately non-intentional facts about how that representation is used (e.g., in neural populations and body mechanics).

Various accounts have been proposed to specify this relation in terms of informational [77,78], causal [79,80], or resemblance relations [81,82,83,84]. The notion of structural similarity falls under the latter approach and is used to elucidate the relation between the internal states of an organism and the external states of its environment in terms of a correspondence function that maps between the relations among the elements of the representational vehicle and its target.

As an early proponent of the structural-resemblance view, R. Shepard assumed ‘second-order isomorphisms’ to specify the relevant mapping, which, accordingly, is such that “the functional relations among objects as imagined must to some degree mirror the functional relations among those same objects as […] directly perceived” [85], p. 131). Backed up by his earlier studies on mental rotation with Jacqueline Metzler [86], Shepard suggested this mirroring relation consists of a bidirectional structural correspondence between distal and proximal relations to explain the agent’s success at solving a task by appeal to a systematic relationship between the external and his assumed internal geometries in psychological space. The correspondence between internal and external states is ‘second-order’ in that it is mediated by first-order invariances in direct perception in the sense of [87]; the correspondence thus concerns relations among relations. In [32], Shepard later refers to this relationship as a ‘psychophysical complementarity’, where he circumscribes it with the metaphor of two structures that relate to each other like a lock and a key. His idea was that only because there is such a complementarity is it possible for internal representations to play an action-guiding role for the agent.

It should be noted that Shepard’s usage of ‘isomorphism’ diverges from its technical definition in abstract algebra. As suggested by Shepard’s lock-and-key metaphor, the term is used analogically, and so the specific algebraic properties required for abstract algebraic isomorphisms need not actually obtain in his geometric model of cognition. Likewise, the way in which the notion is typically used in cognitive science and in the current review is purely analogically; it refers to a mirroring relationship between psychophysical properties. (I am grateful to an anonymous reviewer for pointing me to this distinction).

More recently, [82,84] suggest to refine the relevant structure-preserving mapping in terms of a similarly analogical notion of ‘second-order homomorphism’, which accommodates the idea that the relationship between psychological and physical structures is directional, e.g., in the sense that mental states are about or directed towards physical states (see also Isaac’s [88] account of ‘causally-induced homomorphism’). A remaining worry with the notion is that homomorphisms are cheap and raise triviality worries with the account (anything would count as a representation of anything).

In response, it is typically assumed that mental representations carry only that information that is somehow relevant to guiding the agent’s actions (see [89,90]). The authors of [53] imagine the action-guiding role of probabilistic representations to be played in this way (although they leave open how exactly action-guidance is to be understood formally, that is, under what specific conditions the assumed content in cognitive representations becomes action-guiding). However, while in Shepard’s model, it just so happens that the suitable isomorphism maps geometric structures, from the perspective of structural resemblance approaches more generally, the mapping might as well exist between statistical structures—as seems to be suggested by Lee and Orlandi [53].

The main difference between the notions of structural and geometric similarities is that the latter serves to specify the internal coherence constraints on learning and reasoning, while the notion of structural similarity serves to specify the relationship to external constraints to explain successful behavior (e.g., structure-preserving mappings between an internal map and the external spatial surround explain how a mouse can successfully navigate to a food item in a maze). Representations in conceptual space are purely epistemic entities; the theory makes no commitment to the assumption that concepts represent the world in a truth-conducive manner. In contrast, structural-resemblance approaches additionally consider how internal geometrically structured representations must be arranged to fit the causal structure of the world and guide successful action.

Transferring this perspective to our earlier examples, the explanation of concept learning in this case changes: Learners would be rational to infer Dalmatian (or coral red), as opposed to dog (or red), insofar as structural features associated with these mental representations adequately correspond to invariances in the environment that objectively track relevant distinctions between Dalmatians and dogs. In this case, the content of the concept delivers an explanatory connection with distal features of the agent’s environment.

The reason why ‘fep’ should mean Dalmatian in the earlier example is that the concept appropriately maps onto statistical invariances associated with features of Dalmatians (including applications of the label), while the concept dog inappropriately maps onto the relevant invariances associated with observations of Dalmatians.

Crucially, since structural mappings are cheap, there is an additional requirement to explain why a given behavior (e.g., generalizing words) counts as ‘successful’: what the ‘relevant’ features or invariances are may depend on whether the agent can act on them, that is, whether these invariances are exploitable or ‘there’ for the agent. (There is disagreement concerning whether exploitability acts as a criterion to individuate the content associated with mental representations, i.e., what they are about (e.g., [84]) or whether it is a criterion to identify whether a probabilistic relation counts as representing something at all [90]. In this paper, I stay neutral on this distinction). [84] (p. 10) nicely rephrases this: “Whether a representation is correct or incorrect depends on factors outside the organism, which seem to make no difference to how the representation is processed within the organism (e.g., to how activity of some neurons causes activity of others)”.

Shifting focus on structural similarity adds a possible justification to the assumption that probabilistic representations should be treated as mental entities, insofar as they make, in this sense, ‘exploitable’ predictions about the world. A structure-preserving mapping from objective feature space to internal conceptual space provides representational contents appropriate for the system to use, also in probabilistic learning computations.

While I think that Lee and Orlandi rightly assume a correspondence between subjective probabilities and analog magnitudes of perceptual representation, they do not fully work out how that correspondence comes about. I suggest that we can look at this point to similarity models which fill this gap in several ways in traditional psychological science.

Specifically the analogical notion of structural homomorphism can be used to specify the correspondence relation. Ref. [84] (p. 115) illustrates this with an example from rat navigation: “the discovery of the location-specific sensitivity of place cells does not, by itself, show that rats have a cognitive map. More recent work has shown however that there is an important relation on the place cells, the relation of co-activation. Cells corresponding to nearby locations tend to activate one another.” Shea also discusses the implications of this idea for the problem of content fixation by focusing on the notion of exploitable second-order isomorphism.

Shea classifies his account as falling broadly under the teleosemantics of [91]. Millikan’s approach treats representations as stand-ins that a consumer relies on to deal with an external state of affairs. For instance, consumer honeybees observe wiggle dances of incoming bees as a guide to where nectar is located; they consume this information by reacting on the dance as an intermediary to finding a foraging solution. The consumer subsystem’s behavior counts as “successful” insofar as it carries out evolutionary functions to promote survival and reproduction. However, a major issue with this account is that it remains unclear how we are to identify relevant consumers; this has been shown to be especially difficult in with regards to representations in the brain [92]. Shea’s variation on Millikan’s theme follows up on a suggestion made earlier by [93], to shift focus to the individual learning history, as opposed to the evolutionary learning history when determining the success of the behavior.

Shea argues that structural correspondences are content-constituting if they are exploitable, i.e., can be used by the system and this can explain the system’s performance of task functions. He defines exploitable structural correspondence as “a structural correspondence between relation *V* on vehicles vm in a system *S* and relation *H* on entities $x_{n}$ in which (i) *V* is a relation that processing in *S* is systematically sensitive to; and (ii) *H* and $x_{n}$ are of significance to *S*. Significance to *S* is significance relative to the way outcomes produced by *S* are stabilized and robustly produced” [84] (p. 120). The relativity of significance here means that significance is expressed in a consumer system’s behavioral patterns.

To illustrate this with rat navigation: The relation of co-activation on place cells corresponds to a lot of relations in the world, “but it is the correspondence with the relation of spatial proximity on locations that makes sense of how the animal manages to perform its task functions. Spatial proximity between places is directly relevant to the task of following shortest routes to reward […]” [84] (p. 120). For this reason, cognitive processing is sensitive to the connectivity between place cells, as opposed to their color or where place cells are located in the hippocampus.

Recent work in philosophy of cognition is now considering this idea in the context of predictive processing models of learning via prediction-error-minimization [94]. Gadziejewski’s structural account of PP representations following the second-order homomorphism approach. According to Gadziejewski, the structure of PP models resembles the “causal-probabilistic” structure of the world, where Wiese interprets this to mean that “internal model capture the causal structure of the world; due to noise and partial lack of information (which brings about uncertainty), this model has to be probabilistic; whether the causal structure of the world is itself also probabilistic in some sense is left open” [95] (fn. 6).

The key argument in favor of PP models having cognitive content is that they bear functional relations between variables at different hierarchical levels that structurally map onto causal structure in the world. PP assumes that functional relations between variables in the model are embodied in neural dynamics and that there is a structure-preserving mapping between the world and the brain, in the sense that “variables that can be used to describe processes in the world can be mapped, in a structure-preserving manner, to variables that can be used to describe processes in the brain” (ibid.).

In summary, these are two different ways of constraining probabilistic learning in terms of the notion of psychological similarity.

The first way of understanding similarity as internal representation (e.g., modeled as distances) in psychological space fills out probabilistic predictions with the missing contents but it remains unclear how these contents bear on solving the agent’s task (e.g., inferring the most likely word meaning). In contrast, the second way of understanding similarity as exploitable isomorphism is useful to constrain probabilistic models of learning with cognitive or mental content and shift focus on questions concerning what a specific learning algorithm can do to guide the agent’s selection of actions in a specific environment or ecological niche. That is, the first way provides semantic content to probabilistic computations (answering to what these compute), while the second way constrains them with metaphysical content, answering to the question of which choice of the underlying dimensions to construct semantic content is conducive to successful action.

The two notions reflect principles of computing probabilities that respect their contents (i.e., what probabilities in learning from experience are probabilities of). Both notions find traction in the literature in psychology and philosophy, but only the first way (probabilistic learning as guided by internal coherence norms) has so far received explicit attention in machine learning.

Both notions of similarity deserve further attention as they provide essential constraints on probabilistic models’ usefulness to understanding why and how probabilistic computations result in appropriate behavior, for instance, in learning from experience. Table 1 organizes the different approaches discussed in Section 2 and Section 3 according to the suggested type of content they model and the constraints and norms they place on learning and reasoning.

## 4. Future Work

### 4.1. Placing Probabilistic Constraints on Psychological Similarity Representation

I have argued that several notions of psychological similarity provide important opportunities to constrain psychological explanations of probabilistic learning in mind and machine. One may ask how this proposal stands to authors who have argued that probabilistic norms of belief can be used to constrain assumptions about similarity in perception and experience.

The notion of psychological similarity has been put off (too quickly from my point of view) as scientifically useless by many, most famously perhaps by Nelson [96], who characterizes it as an “imposter” and a “quack”. Goodman’s worries with the notion mainly targeted issues of trivialization: anything (e.g., my foot) can be interpreted to be similar to anything else (e.g., my desk) in some respect (e.g., with regards to their distance to the moon). It also seemed as if one could not escape trivialization by arguing that the relevant respects could be rationally inferred, for if such inference itself was to be explained by similarity, this would lead to a vicious circle (see [97] for discussion).

For this reason, many cognitive psychologists such as [2] explicitly attempt to rely on “a measure of similarity that is free from *contamination* by inferential reasoning” since, so they claim, the “measure of similarity is directly mediated by the mental representation of concepts (i.e., the representations of an object *o* as belonging to a category *C*) rather than accessing the machinery of inductive cognition” [2] (p. 261, emphasis added). On several theories of concepts, the contents of concepts are themselves constituted by inferential relations, a version of inferential-role semantics and theory-theories of concepts [98,99]. Thus, just showing that the measure of similarity is mediated by concepts is insufficient to show that similarity representations are not “contaminated” by inductive inference during probabilistic learning.

Overall, the debate remains inconclusive, but progress has recently been made. For instance, one recent proposal is that probabilistic biases are active in selecting the relevant dimensions to initially construct similarity space and select relevant features to assess what respects and contrast classes are relevant in a given context of similarity judgment [100]. Given assumptions about the task (e.g., navigation) and information about the environment statistics, an agent infers how mutually informationally relevant dimensions are (e.g., color or hue is intuitively informationally irrelevant to location, while hue saturation and brightness are highly mutually relevant). Sets of dimensions may then be selected by choosing the set of dimensions that appears to be most mutually informationally relevant.

To avoid worries concerning circularity, let me point to Gärdenfors’s assumption that the initial construction of low-dimensional space results from associationist mechanisms that outsource the correlations found in high-dimensional subsymbolic space. Once such low-level connections are drawn, probabilistic inference may place useful constraints on the selection of relevant domains at the conceptual level of representation. The assumption of layered representational kinds (i.e., subsymbolic, conceptual and symbolic) allows us to avoids a vicious regress, since the dimensions themselves are not derived from probabilistic inference.

Nevertheless, it remains currently unclear whether it would be coherent to assume that even the selection of dimensions to constitute a similarity space could depend on higher-order beliefs; what contrast classes are used to carve up similarity space into concepts may also be influenced by top-down probabilistic inferences at higher levels (e.g., in symbolic space).

### 4.2. Drawing Conceptual Distinctions between Psychological Similarities and Probabilistic Dependence

It is also important to note that the notions of psychological similarity and probabilistic dependence need to remain clearly conceptually distinguished (I am grateful to Karin Enflo for raising my awareness to this point). That the notion of probabilistic dependence cannot replace the notion of similarity can be illustrated with a toy example.

You derive the similarity between a raven and a desk based on identifying the probabilistic dependence of the event of a raven occurring and the event of a desk occurring. The higher this mutual probabilistic dependence relation, the more similar the raven and the desk will be. (This is parallel to how one may think about connectionist networks drawing associations between network nodes representing desks and ravens). We assume that opposite things should have a low score of similarity, but this does not imply that they are mutually irrelevant. Now the counterintuitive result is that the negation of the proposition “there is a raven” (¬P) is maximally informationally relevant to the positive hypothesis “there is a raven” (P) but it is also maximally dissimilar to it. In fact, ¬P and P are opposites in terms of their truth-values.

This shows that our intuitions about similarity can come apart from our intuitions about probabilistic dependence. Furthermore, it is impossible to find any structure-preserving mapping between these two statements since they are contradictory with respect to their logical and semantic structure.

This situation is complicated by the fact that psychological similarity obtains multiple meanings, as illustrated by the contrast between geometric and structural similarity in Section 3. These multiple meanings need to yet be made formally precise. One possibility, suggested by an anonymous reviewer, is to explore whether the notions map onto the mathematics of manifolds, as a way to formally specify geometric similarity, versus the mathematics of discrete topological spaces, to formally specify structural similarity.

### 4.3. Refining Theoretical Connections between Psychological Similarity and Probabilistic Learning

Finally, it also needs to be elucidated further what principles may connect the two ways of understanding probabilistic learning based on psychological similarity or how they could be unified into a single framework.

One early but not fully successful attempt to this task was made by Roger [32,101], who suggested that agents who represent the world in terms of an internal measure of similarity achieve successful categorization into useful kinds (useful in the sense that the agent’s own internal representational construction of these kinds is conducive to guiding action and control of its environments to satisfy the agent’s needs and goals). He assumed second-order isomorphisms, on the grounds of his studies on mental rotation with Jacqueline Metzler [102] to independently motivate the similarity-spaces framework, based on which he derived the universal law of generalization using a Bayesian inference scheme.

According to the similarity-spaces framework, what probabilities are relevant to decide on an action depends on how similarity space is carved up [60]. On this view, probabilistic computations obtain their action-guidance and ground their accuracy in experience by exploiting similarity spaces as an intermediary. This short overview illustrates that further work is needed to understand the theoretical relationships between probabilistic learning and the various notions of psychological similarities. Thus, future research should focus on examining the applicability and implications of combining psychological similarity spaces and probabilistic learning.

### 4.4. Connecting Psychological Similarity and Neural Representations

Another relevant avenue for future work is a closer examination of how the outlined geometric and structural similarity approaches to specifying probabilistic content relate to the literature on neural representations in biological and artificial networks. As explained in Section 3.1, Ref. [54] takes the notion of geometric similarity representation to be most useful to describe cognitive representations at the conceptual level, as opposed to the subsymbolic and symbolic levels of cognitive representation.

The geometric-similarity account can be understood as a proposal concerning the right level of description for psychological explanations of how cognitive representations support flexible behavior. As a methodological bridging tool (Section 3.1), the theory of conceptual spaces suggests that the geometric level of description is specifically well suited for analyzing behavioral regularities, including neural activity patterns, that seem to be involved in representing stable, long-term regularities associated with categorizations of objects that fall under the same kinds.

Much research in contemporary computational neuroscience now also suggests a stronger reliance on geometric structure when analyzing representations in biological brains and deep neural networks [103,104,105]. The re-use of geometric terminology across these scientific fields suggests a potential avenue to connect psychological explanations of high-level cognition (e.g., conceptual thought) to neuroscientific explanations of neuronal information processing (e.g., about spatial distances).

Nevertheless, it is important to note that making this connection requires first relating two key features associated with mental representations: (a) their physical realization in neural vehicles, which provides representations causal powers to produce behavior and which is the primary focus of systems neuroscience; and (b) the intentional status of mental representations, i.e., their meaning or content, which explains why the behavior was produced and which is the primary focus of similarity-based approaches.

However, there remains little work on the role of structural similarities in drawing comparisons between the brain activation patterns of different animals and comparisons between artificial neural networks and biological brains. Recent work by [106] provides a starting point for an investigation of this issue in neural-network modeling in systems neuroscience. They develop a set of criteria to identify an appropriate model-to-mechanism mapping in systems neuroscience, a notion initially introduced by [107].

According to Cao and Yamins, the relevant criteria “require us, first, to identify a level of description that is both abstract but detailed enough to be ‘runnable’ [by which they mean that this description maintains ‘those features that are causally sufficient to generate the phenomenon of interest’], and then, to construct model-to-brain mappings using the same principles as those employed for brain-to brain mapping across individuals” [106] (p. 1). To identify the appropriate mapping, [106] (p. 1) focus on “assessing the functional similarity of a model to its explanatory target” [106] (p. 4, my emphasis). As a “reasonable first pass” [106] (p. 19), Cao and Yamins suggest assessing the adequacy of a proposed model-to-brain mapping (or, analogously, the brain-to-brain mapping across individuals) in terms of their notion of a “similarity transform”.

Accordingly, sets of neurons in corresponding brain areas in two animals (or in an animal and a model, for that matter) “are similar when one [set] can be constructed from the other via an invertible linear transform’’, such that one animal is taken “to be a ‘linear model’ for the other” [106] (p. 18). When a similarity transform exists, and the individuals’ (or the model’s and the individual’s, for that matter) “neural responses in corresponding brain areas line up under linear transform”, then the “two individuals are said to be equivalent […], this means that the individual’s performance on any behavioral task (assuming the brain areas involved are actually being used to support that task behaviorally) will be the same” [106] (p. 18).

For Cao and Yamins, the notion of similarity transform captures the right level of abstraction for mechanistic explanations in neuroscience because it allows one to “quotient out’ the inter-animal differences by means of the proper class of inter-animal transforms, creating a ‘population equivalence class’ as the target of explanation” [106] (p. 19). This allows neuroscientists to generalize relevant properties across activity patterns of individuals to trace the causally relevant factors in the production of the corresponding type of behavior.

However, note that Cao and Yamins, in focusing on relations between neural activity patterns, focus on features of representational vehicles. (The same focus arises for alternative methods to compare neural activity patterns such as Representational Similarity Analysis, which has been proposed by [108] to methodologically unify systems neuroscience). From the mechanistic point of view, these features purely depend on physical aspects of the system, its parts, activities, and their organization. It remains open what and why these neuronal populations represent what they do (see [84] pp. 29–30, for a philosophical discussion).

Structural-similarity accounts complement vehicle-focused neuroscientific explanations with content-focused psychological explanations. That such a complement is useful is clarified by [84] (p. 7), who writes:

The lack of an answer to the content question does arouse suspicion that mental representation is a dubious concept. Some want to eliminate the notion of representational content from our theorizing entirely, perhaps replacing it with a purely neural account of behavioural mechanisms. If that were right, it would radically revise our conception of ourselves as reason-guided agents since reasons are mental contents. […] But even neuroscientists should want to hold onto the idea of representation, because their explanations would be seriously impoverished without it. Even when the causes of behaviour can be picked out in neural terms, our understanding of why that pattern of neural activity produces this kind of behaviour depends crucially on neural activity being about things in the organism’s environment.

In other words, explanations in neuroscience should not only seek to record and relate activity patterns in to-be-compared neural areas (as is suggested by Cao and Yamins’ ‘first pass’ reliance on similarity transforms), but additionally a theory is needed to explain “how the activity of those areas should be understood as representing things about the stimuli presented to the people doing a task” [106] (p. 8).

Structural similarity accounts attempt to fill this gap and explain how “[t]he content of a neural representation makes an explanatory connection with distal features of the agent’s environment, features that the agent reacts to and then acts on” [84] (p. 8). That is, structural similarity accounts offer a potential way to supplant neuroscientific explanations “with further information about how neural properties relate to worldly properties [so that] the correlations with the worldly properties would be doing all the explanatory work” [84] (p. 103).

On teleofunctional views, such information would be delivered by etiological functions that determine the sense in which a representation is used by a consumer system as a stand-in for a target [76]. Although structural-similarity approaches to mental representation have recently received attention from proponents of mechanistic cognitive science (e.g., [109]), the role of the notion in mechanistic accounts of how mental representations might be instantiated in physical systems and explained by nonintentional facts about the world remains insufficiently explored.

New mechanists rather tend to focus on the teleosemanticist’s key notion of ‘proper function’ (see, e.g., [110,111,112]). They commonly assume that representational contents depend not only on the way in which vehicles are related to their targets but also on the function that these representations play within the cognitive system more generally. One way to fix content is by looking at aetiological functions, where a representation is used by a consumer system as a stand-in for a target [76]. For example, the biological function of a mammal’s heart is to pump blood because circulating blood is the effect for which hearts have been evolutionarily selected for. This leaves future work with the task of clarifying whether and to what extent adding such information satisfies new mechanistic explanatory criteria such as model-to-mechanism mappings.

## 5. Conclusions

This opinionated review has critically evaluated the explanatory contributions of probabilistic models with a focus on concept learning from limited data in minds and machines. I have focused on one key worry, namely, that probabilistic computations lack cognitive content, and argued that this gap can be filled by deferring to the notion of psychological similarity.

I have discussed how two distinct notions of psychological similarity—distance in psychological space and structural resemblance—can inform this project. In particular, the two notions furnish probabilistic models of cognition with meaningful interpretations of what the associated subjective probabilities in the model represent and how they attach to experiences from which the agent learns. Similarity representations thereby provide probabilistic models with cognitive, as opposed to purely mathematical, content.

Future research should elucidate (1) how probabilistic assumptions can place fruitful constraints psychological similarity representations while (2) keeping a clear conceptual distinction between the two notions, and (3) should refine possible theoretical connections between similarity-based and probabilistic approaches to learning.

## Figures and Tables

**Table 1 entropy-25-01407-t001:** Organization of probabilistic, geometric, and structural similarity approaches to specifying semantic content in probabilistic hypotheses spaces. The specification is according to three categories: the type of content these models represent (i.e., mathematical/probabilistic or cognitive/intentional) and the constraints and inference norms they place on learning and reasoning (e.g., internal or external norms).

Content-Type	Constraints	Norms	Approach
Cognitive	Second-order isomorphism, action-guidance	Coherence with external geometries	[32]
Mathematical	Implicit assumptions about the data-generating (sampling) process	Internal coherence with probability axioms	[6,12,13,14]
Cognitive	Cognitive design principles	Internal coherence with geometric axioms	[5,54]
Cognitive	Second-order homomorphisms, action-guidance	Coherence with external environment statistics	[53,84]

## Data Availability

No new data were created or analyzed in this research. Data sharing is not applicable to this article.

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
