# Peer review of "Probabilistic Learning and Psychological Similarity"

_entropy, 2023, doi:10.3390/e25101407_

Round 1
Reviewer 1 Report
This paper presents an analysis of probabilistic models of concept learning from the perspective of the philosophical literature. The primary concern is that such models specify how beliefs about concepts should be updated in light of data, but do not make it clear what the content of those concepts are or how they relate to experience. Some solutions to this problem based on previous work in philosophy are suggested, and the paper highlights directions for future work.
I'm not a philosopher and can't evaluate the philosophical arguments or engagement with that literature, so I will focus my comments on the interpretation of the models of concept learning. To make this straightforward I will go page by page:
p 3. I don't think the references to Lake et al. are relevant to the model being presented, which originated in Tenenbaum (1999) and Tenenbaum (2000) and was elaborated in Tenenbaum and Griffiths (2001) and Xu and Tenenbaum (2007). There are also earlier rational models of concept learning in Shepard (1987) and Anderson (1990) that are probably worth citing.
I think footnote 2 should probably appear in some form in the main text, as this was the primary motivation for this model of concept learning.
p 6. The critique presented here seems to treat the model as primarily being the belief updating process, but the specification of the hypothesis space is just as important. Indeed, this is what generates the predictions of the models. In all of the models discussed, concepts are defined extensionally -- they are sets of objects. This is what specifies what the concepts are, and the prior distribution specifies the results of past experience in shaping expectations about these concepts. I think disregarding the importance of the hypothesis space is what leads to this critique.
It's also worth noting that in later version of these models the hypothesis space is elaborated. In Xu and Tenenbaum it is a hierarchy that is used in defining the hypotheses, and in subsequent more recent work representations such as logical formulas and programs are used to generate hypotheses (and to provide them with intensional interpretations).
p 8. I found section 3.1 confusing, as the "regions of psychological space" idea is exactly how concepts are defined in Shepard (1987), and in turn inherited in some of the models presented in Tenenbaum (1999) and Tenenbaum and Griffiths (2001). It would seem that this solution has already been pursued.
I think addressing these points is important before this work is ready for publication.
The paper is generally well-written but given the interdisciplinary audience of this special issue I think it's worth investing additional effort to define ideas from the philosophical literature that may not be transparent or could be interpreted in a non-technical way, and that these should be supported through further more basic references for outsiders to philosophy who want to dig deeper.
Reviewer 2 Report
Summary
This paper presents a well-written, critical review on the frameworks of psychological similarity and probabilistic learning within the context of concept learning in computational cognitive science. The author notes that probabilistic learning approaches (typically rooted in Bayesian inference) are able to provide a normative account for how, algorithmically, information should be combined to make inferences, while being agnostic to what that information is. However, the structure of information heavily influences inferences. For example, we can generalize inference between contexts based on the similarity of one context to another. This lack of focus on representation is an important problem in the domain of probabilistic modeling in cogsci. This point is summarized nicely by the author: "[T]hese issues call for a stronger focus on the semantics of probabilistic inferences rather than on the computational principles based on which they may be processed in the mind."
The author suggests that the framework of psychological similarity should be combined with the probabilistic approaches to supply the content to the structure. The author reviews recent works that do this, using a traditional perspective on psychological similarity, in terms of continuous distances in a psychological representation space: "What is common to all these approaches is that, to the extent that they explain behavioural patterns that illustrate probabilistic forms of inductive inference (e.g., perceptual colour categorisation), they stipulate that the relevant inferences range over areas in psychological similarity space, and that it is the geometric structure of that space which grounds probabilistic computation in representational content and experience. In particular, how probabilities are assigned may depend on what dimensions are salient and also which dimensions are combined into domains." The author also reviews papers that supply the content to probabilistic inference according to structural similarities -- "exploitable isomorphisms" between contexts or concepts. The author notes that the first case has been received more attention in machine learning than the second.
Overall, I believe the review presents an interesting perspective and useful synthesis of two important lines of work in computational cognitive science. The author puts the two lines of work into a common perspective, highlighting the distinction between the more "syntactic" approach of probabilistic modeling and "semantic" approach of psychological similarity.
Suggestions
I recommend a few minor revisions:
The section on structural similarity would benefit from some revision to make the concept of structural similarity -- and how it differs from geometric similarity -- more clear at the outset. In addition, this distinction could be made earlier in the paper. It is alluded to, but the author could make it more clear and explicit, and highlight it as a clear contribution of the review.
In addition, some paragraphs throughout the paper are excessively long, and should be broken up into smaller units.
I would also recommend that the author consider including a master figure that organizes the main approaches from the literature according to the axes along which she considers them, which will help the reader quickly get the gist and will be a useful reference. This is not necessary, but if it is possible to make a clear and visually intuitive figure like this, it would be a nice boost.
Considerations for Future Work
The author highlights two important notions of similarity: "geometric" and "structural." While not needed for this review, I believe these two categories could be more formally fleshed out in terms of the mathematics of manifolds vs. discrete topological spaces -- this might be an interesting idea to explore in future work.
In future work, it may also be useful to consider the relationship between this literature and the literature on neural representations in both biological and artificial neural networks. There's extensive work comparing neural representations in brains and machines (e.g. the work of Dan Yamins, et al), as well as analyzing the geometric structure of concept space in brains and deep networks (e.g. Alexander Huth, Alex Williams). The "psychological similarity" (i.e. representational distance) idea has been heavily explored in this field. However, little work has been done on the notion of "structural similarity" between neural representations, and it would be useful to highlight and motivate the potential for this type of analysis.
Round 2
Reviewer 1 Report
This revision is significantly improved from the previous version and the argument is clarified. I think this argument will primarily be of interest to philosophers, but the paper is more accessible to readers from other disciplines.
There are some minor writing issues that could be addressed:
p 5. “their exclusive reliance” is ambiguous as to whether it refers to T&G or previous models
”one such a bias” -> “one such bias” or “such a bias”
p 8. Lake et al (2017) is presumably (2015)?
p9. “critique on” -> “critique of”
p 11. “For instance, any instance”
Author Response
NB: All changes to the manuscript are highlighted in blue color. The formatting has been changed according to the MDPI template.
Point 1: There are some minor writing issues that could be addressed:
- p 5. “their exclusive reliance” is ambiguous as to whether it refers to T&G or previous models
- ”one such a bias” -> “one such bias” or “such a bias”
- p 8. Lake et al (2017) is presumably (2015)?
- “critique on” -> “critique of”
- p 11. “For instance, any instance”
Response 1: Thank you, I have made the corresponding changes to address these writing issues, point by point:
- I have changed this to “These earlier attempts failed …” on p. 4 of the revised manuscript.
- I have changed this to “one such bias” on p. 4 of the revised manuscript.
- Yes, thank you for highlighting this – I have changed this to “Lake et al.’s [2015]” on p. 6 of the revised manuscript.
- I have corrected this on p. 7 of the revised manuscript.
- I have replaced “instance” with “example” on p. 9 of the revised manuscript.

Reviewer 2 Report
The edits made by the author provide many improvements to the paper, including a nice new section that draws connections to the systems neuroscience literature. The paper is suitable to be published in its current form. However, I would recommend two considerations for the author before submitting the final draft:
1. There are still many long paragraphs that could be broken up into shorter sections. I recommend a final pass of editing for concision and breaking up long sections.
2. I would be careful with the use of the words "isomorphism" and "homomorphism." These words have precise technical definitions from abstract algebra which require the structures over which the mapping is made to satisfy certain algebraic properties that generally do not hold in most cognitive models. I believe the author is using these terms more analogically. I think the analogy is good. I would recommend making it explicit that the usage is analogical -- or coming up with two novel terms to take on the same meanings.
Author Response
NB: All changes to the manuscript are highlighted in blue color. The formatting has been changed according to the MDPI template.
Point 1: There are still many long paragraphs that could be broken up into shorter sections. I recommend a final pass of editing for concision and breaking up long sections.
Response 1: I have carefully edited the revised manuscript to shorten any lengthy paragraphs throughout the manuscript.
Point 2: I would be careful with the use of the words "isomorphism" and "homomorphism." These words have precise technical definitions from abstract algebra which require the structures over which the mapping is made to satisfy certain algebraic properties that generally do not hold in most cognitive models. I believe the author is using these terms more analogically. I think the analogy is good. I would recommend making it explicit that the usage is analogical -- or coming up with two novel terms to take on the same meanings.
Response 2: Thank you for highlighting this important distinction. Yes, I use the notions analogically in the paper, akin to how R. Shepard had used the notion of second-order isomorphism to motivate his geometric model of cognition. Shepard initially referred to the relationship as one of “mirroring” or “psychophysical complementarity” and relied on the metaphor of a lock and a key to illustrate the analogical meaning. I have explicitly clarified on p. 13 of the revised manuscript that the intended meaning of these terms in cognitive models is analogically.
